# Amphiphilic Grafted Polymers Based on Citric Acid and Aniline Used to Enhance the Antifouling and Permeability Properties of PES Membranes

**DOI:** 10.3390/molecules28041936

**Published:** 2023-02-17

**Authors:** Jiahui Zhao, Peng Zhang, Lin Cao, Haoling Huo, Huaijun Lin, Qiwei Wang, Florian Vogel, Wei Li, Zhidan Lin

**Affiliations:** Institute of Advanced Wear & Corrosion Resistant and Functional Materials, Jinan University, Guangzhou 510632, China

**Keywords:** water treatment, ultrafiltration membrane, amphiphilic polymer, antifouling, permeability

## Abstract

Water treatment technology based on ultrafiltration (UF) faces the problem of severe membrane fouling due to its inherent hydrophobicity. The use of amphiphilic polymers that possess both hydrophobic and hydrophilic chain segments can be advantageous for the hydrophilic modification of UF membranes due to their excellent combination in the membrane matrix. In the present study, we examined a novel amphiphilic CA–g–AN material, constructed by grafting citric acid (CA) to aniline (AN), as a modified material to improve the hydrophilicity of a PES membrane. This material was more compatible with the polymer membrane matrix than a pure hydrophilic modified material. The polyethersulfone (PES) membranes modified by amphiphilic CA–g–AN demonstrated a higher water flux (290.13 L·m^−2^·h^−1^), which was more than eight times higher than that of the pure PES membrane. Furthermore, the flux recovery ratio (*FRR*) of the modified membrane could reach 83.24% and the value of the water contact angle (WCA) was 76.43°, demonstrating the enhanced hydrophilicity and antifouling ability of the modified membranes. With this study, we aimed to develop a new amphiphilic polymer to improve the antifouling property and permeability of polymer-based UF membranes to remove organic pollutants from water.

## 1. Introduction

The conversion of wastewater into clean, renewable water is critical technology for the development of a sustainable energy economy in the future [1,2,3]. To date, a major challenge for the large-scale separation and purification of wastewater remains the lack of low-cost, highly efficient, and conventionally applicable wastewater treatment technology. Membrane separation technology has seen considerable progress in recent years, even though water treatment technologies such as adsorption [4,5], flocculation [6,7], ion exchange [8,9], and distillation [10] have been commercialized for decades. In addition to judiciously removing multiple impurities, it is simple and fast to operate and requires no or minimal chemical additions in the application process. Emerging membrane separation technology has attracted considerable attention due to the higher efficiency, lower energy consumption, and fewer demands of time and space investments than conventional technology [11,12,13]. In addition, membrane separation technology can be classified as microfiltration (MF), ultrafiltration (UF), nanofiltration (NF), and reverse osmosis (RO).

In the case of UF technology, pollutants can be efficiently and effectively removed through the pressurization and physical screening of the key components of the membrane. As a result of its excellent chemical stability and acid and alkali resistance as well as its mechanical properties, polyethersulfone (PES) is one of the most commonly used materials [14]. The problem with PES UF membranes is that, due to their inherent hydrophobicity, they suffer from problems relating to membrane contamination and pore blockage, which largely limit their long-term durability [15,16]. However, PES membranes are also known for a decrease in their penetration during the application process. To address the above aspects, an efficient and practical method of hydrophilic membrane modification should be developed to produce membranes that are antifouling and highly permeabilized under practical conditions.

Traditional treatments for overcoming the hydrophobic characteristics of PES membranes include surface coating [17], surface grafting [18], physical adsorption [19], blending with hydrophilic polymers [20], plasma treatments [21], or the UV-induced grafting of hydrophilic functional groups [22]. A combination of hydrophilic polymers such as polyvinylpyrrolidone (PVP) [23], polyethylene glycol (PEG) [24], and polyvinyl alcohol (PVA) in the polymer matrix membrane is the most widely used and easiest strategy to modify PES membranes. Nevertheless, as the above-mentioned hydrophilic polymers are water-soluble, there is an inevitable elution of modified materials that have been exposed to water for an extended period of time [25,26,27]. Thus, amphiphilic polymers have attracted a great deal of interest due to their combination of hydrophilic and hydrophobic chains [28,29,30]. As a result of the hydrophobic chains present on amphiphilic polymers [31], they are well-compatible with polymer matrix chains; the hydrophilic part of amphiphilic polymers can stably modify the inner and outer surfaces of membranes through hydrophilic functional groups on the amphiphilic polymers, improving the antifouling properties, hydrophilic properties, and permeability of the membranes. According to Wang et al. [32], hydrophilic polyvinylpyrrolidone (PVP) was grafted onto hydrophobic polyvinyl chloride (PVC) to synthesis an amphiphilic graft polymer (PVC-g-PVP) by atom transfer radical polymerization (ATRP). The incorporated amphiphilic PVC-g-PVP composite membranes demonstrated significant antifouling properties. In another study, Wu et al. [33] fabricated a PVC-g-PEGMA copolymer, also via ATRP. As a result of the modification of the PVC-g-PEGMA copolymer, the membranes obtained excellent hydrophilicity. Mahdavi et al. [34] used polyvinylidene difluoride (PVDF) and polymethyl methacrylate (PVDF) to prepare a PVDF-g-PMMA copolymer by ATRP. The amphiphilic polymer PVDF-g-PMMA was combined with inorganic nanoparticles to modify the PVDF membrane; the resulting composite membrane had a good fouling resistance and high dye removal efficiency. Although there have been recent advances in the development of amphiphilic graft polymers that enhance the antifouling properties and high permeability of PES membranes, existing amphiphilic graft polymers produced by ATRP are still necessary for the precise design of the structures of amphiphilic graft polymers. It is necessary for the ATRP synthesis process to have an oxygen-free environment or high temperatures, which increase the unnecessary cost of the process [25]. Furthermore, the precise structure of amphiphilic graft polymers is largely determined by controlled/free-radical polymerization, narrowing the range of materials that can be modified through amphiphilic graft polymerization [35].

Here, for the first time, we report an amphiphilic graft polymer (CA–g–AN) based on hydrophilic citric acid (CA) and hydrophobic aniline (AN) as a new type of modifier of a PES membrane. By activating the citrate carboxyl groups in CA and grafting them with AN, the amphiphilic graft polymer was simply synthesized by amide and hydrogen bonds. Different from most modified membranes, which are fabricated by conventional blending methods, the CA–g–AN modified membranes demonstrated a higher permeability of water and more stable antifouling property. The water flux of the membrane modified by CA–g–AN reached 290.13 L·m^−2^·h^−1^, and the BSA rejection remained at a high rate of 97.36%. In addition, the antifouling property of the modified membrane was also improved and the flux recovery ratio (*FRR*) reached 86.26%. The amphiphilic CA–g–AN provided the possibility of fabricating the amphiphilic graft polymer via a simple amide grafting reaction, consequently broadening the applications of PES ultrafiltration membranes in a wide range of wastewater treatment fields.

## 2. Results and Discussion

### 2.1. Characterization of CA–g–AN

#### 2.1.1. ^1^H NMR Analysis of CA–g–AN

After the activation of the carboxyl group on CA, the activated carboxyl group reacted with the amino group in AN. As a result, there were five possibilities for the final product (Figure 1). Therefore, the final product could be a mixture. After purification, it was found that AN and CA were grafted by ^1^H NMR. Figure 2 demonstrates the ^1^H NMR spectra of CA–g–AN in DMSO-d6. The chemical shifts at 2.76 and 2.24 ppm were attributed to –CH_2_– (H–e and H–e’); however, the chemical shifts at 6.88, 7.28, and 7.51 ppm belonged to H (H–d, H–c, and H–b) in AN whereas the chemical shift at 9.98 ppm belonged to H–a in AN. ^1^H NMR indicated that CA and AN had been successfully grafted.

#### 2.1.2. FT-IR Analysis of CA–g–AN

FT-IR was used to analyze the chemical structure of CA–g–AN. As shown in Figure 3a, the characteristic vibrations peaks of –OH, –C=O, and –CH_2_ from CA were observed at 3307 cm^−1^, 1635 cm^−1^, and 2631 cm^−1^, respectively. According to the FT-IR spectrum of CA, the characteristic peak at 1732 cm^−1^ may have been related to the vibrational coupling of the two C=O bonds in CA whereas the characteristic peaks at 1397 cm^−1^ and 1227 cm^−1^ could be attributed to the coupling effect of the C–OH stretching vibration and bending vibration. It transpired that CA contained a carboxyl group. However, after grafting AN onto CA, there was an apparent change in the spectrum of CA–g–AN. Compared with the pristine CA, CA–g–AN showed new peaks at 1738 cm^−1^, 1540 cm^−1^, and 1249 cm^−1^. The three peaks were attributed to the amide reactions between CA and AN, and were classified as amide I peak, amide II peak, and amide III peak [36]. In addition, the curve of CA–g–AN showed a peak at 3447 cm^−1^, which represented the stretching vibration of –OH, whereas a sharp peak at 3319 cm^−1^ indicated the hydrogen bond, which occurred when the amino group of aniline interacted with the hydroxyl group and the carboxyl group. The spectrum of CA–g–AN also exhibited a characteristic peak of the benzene ring; out-of-plane bending vibration peaks of =C–H on the benzene ring appeared at 752 cm^−1^ and 692 cm^−1^. As benzene rings only occur in AN, the presence of the characteristic peaks of benzene rings in the spectrum of CA–g–AN further indicated that CA and AN had been successfully grafted.

#### 2.1.3. TG Analysis of CA–g–AN

An investigation of the thermogravimetric performance of the two samples, CA and CA–g–AN, was conducted (Figure 3b). According to the TG of CA, there are three stages of weight loss. Initially, water molecules are released below 100 °C; then, between 100 and 173 °C, labile oxygen-containing functional groups such as –OH, –CO, and –COO are pyrolyzed [37]. The last step occurs from 173 °C to 299 °C, followed by the thermal degradation of the organic backbone of CA [38]. After 300 °C, no weight change is observed in CA. An amide bond was used to graft aniline onto CA, thus enhancing the thermal stability of CA–g–AN. The thermal decomposition of CA–g–AN followed a four-step process. First, the temperature rose from 35 °C to approximately 100 °C, resulting in a weight loss of about 5.15 wt% of water. During the second step, from 100 °C to 199 °C, oxygen-containing functional groups contributed to the breakdown of the amide bonds. It should be noted that the third step, which involved the loss of an amino group in the amide bond as well as the thermal degradation of the organic backbone of CA, occurred between 197 °C and 251 °C [39]. Finally, the slight degradation between 251 °C and 800 °C was related to the decomposition of the aniline skeleton of CA–g–AN. In Figure 3b, the DSC curve shows a trend of heat absorption and release that was consistent with the TG test. Accordingly, the TG analysis confirmed that the thermal stability of the amphiphilic polymer obtained by grafting was further improved as well as the successful synthesis of the graft polymer CA–g–AN.

#### 2.1.4. XPS Analysis of CA–g–AN

An XPS analysis was conducted to further characterize the chemical composition and valence states of the prepared materials. In Figure 4a, the survey spectrum indicates that carbon (C) and oxygen (O) elements were present in CA; nitrogen (N) appeared in CA–g–AN following the grafting of AN. As can be seen in Figure 4b, the deconvoluted C 1s spectrum of CA exhibited three main peaks at 285.04 eV, 286.94 eV, and 289.14 eV, corresponding with C–C, C–O, and C=O, respectively [40,41,42]. Regarding CA–g–AN in Figure 4c, during the analysis of the N 1s spectrum, the peak at 399.24 eV could be primarily attributed to the C−N bonds that originated from the amide bond between CA and AN, thus confirming that CA–g–AN was successfully synthesized. According to the XPS spectrum of C 1s (Figure 4b), the peak of C=O in CA–g–AN shifted to a lower binding energy compared with CA (from 289.14 eV to 287.88 eV), confirming the strong hydrogen bonding interaction between the hydroxyl, amino, and carboxyl groups of CA as well as the amino group of AN [43]. Additionally, there was a new peak located at 285.79 eV, which represented the C–N bond [44]. The XPS analysis further demonstrated that CA and AN had grafted together.

### 2.2. Hydrophilicity of the Membranes

In order to evaluate the surface properties of membranes, it is necessary to measure their water contact angle (WCA). As shown in Figure 5a, the WCA value of the pure PES membrane was 89.26 ± 1.86°, which was mainly attributed to the inherent hydrophobicity of the polymer membrane matrix. However, the addition of CA–g–AN could result in a decrease in the WCA value of the membranes modified by CA–g–AN due to the presence of a large number of hydrophilic functional groups (–COOH and –OH) in CA that increased the level of water interaction [45]. In the presence of 0.25 wt% CA–g–AN, the WCA value reached its lowest level (76.43 ± 1.71°). In light of the reduced WCA value of the modified membranes, CA–g–AN appeared to have a hydrophilic modification effect on the PES membranes. The enhanced hydrophilicity could reduce the possibility of organic protein adhesion, enhancing the antifouling property of the membrane.

The antifouling mechanism diagram of the membranes is shown in Figure 5b. As a result of the hydrophobic properties of PES membranes, organic foulants can easily adhere to the surface of the membrane and aggregate. Organic foulants accumulate on membrane surfaces and around membrane pores, compromising the rejection effect and reducing the reuse performance of the membrane, resulting in economic losses and shortening its service life. To improve the hydrophilicity of a pure PES membrane, it is highly desirable to modify it. CA–g–AN was an amphiphilic graft polymer that consisted of hydrophilic (–COOH and –OH from CA) and hydrophobic (benzene ring from AN) chain segments. In the case of PES membranes, AN may serve as a “fixation site” for hydrophilic functional groups (–COOH and –OH) on CA to adhere to the surface and pores of the membrane, thereby improving its hydrophilicity. As a result of the fixed hydrophilic functional groups, the modified membranes were able to repel hydrophobic foulants, thereby improving the antifouling property, reusability, and permeability of the PES membrane.

### 2.3. AFM Observation and XPS Analysis of the Membrane Surface

Figure 5c shows the AFM images and surface roughness values of the prepared membranes. In comparison with a pure PES membrane, the membrane modified by 0.25 wt% CA–g–AN demonstrated a lower roughness in accordance with the results of the WCA tests. The reason was that the modification additive, amphiphilic CA–g–AN, was more compatible with the membrane matrix in the blending process, which reduced the roughness of the membrane surfaces, promoting a smoother membrane surface. Furthermore, the XPS analysis of PES/0.25–CA–g–AN is presented in Figure 5d. Based on the results of the XPS test, it was determined that the elements C, N, O, and S were uniformly distributed on the membrane, further indicating that the modified material CA–g–AN was strongly attached to the surface of the membrane.

### 2.4. Membrane Morphology, Porosity, and Mean Pore Size Analysis of the Membranes

As shown in Figure 6a, SEM was conducted in order to understand the modified effect of CA–g–AN on the membrane microstructure. The morphology of the pure PES membranes was irregular and the bottom surface contained a number of macro-voids. In contrast, the membranes modified by CA–g–AN exhibited a more regular cross-sectional microstructure compared with the pure PES membrane. When the concentration of CA–g–AN in the coagulation bath increased from 0.10 wt% to 0.25 wt%, the membrane pores increased in size, length, connectivity, and uniformity, resulting in less resistance to water transport across the membrane in comparison with the small and short finger-like pores. As a result of the hydrophilicity groups (–COOH, –OH, and amide groups) of CA on CA–g–AN, the pore structure of the modified membranes may have been altered, affecting the exchange rate between the solvent and non-solvent (water) [46]. The distribution of the C, N, O, and S elements on the PES/0.25–CA–g–AN membrane is illustrated in Figure 6b.

A comparison of the porosity and mean pore size of the membranes is presented in Table 1. The pure PES membranes had a porosity of 66.84% and a mean pore size of 47.07 nm, respectively. It was observed that with the improvement in the concentration of CA–g–AN, both the porosity and mean pore size increased after the modification by CA–g–AN. Improved pore structures can improve membrane permeability. As the CA–g–AN concentration reached 0.25 wt%, the porosity and mean pore size reached their maximum levels.

### 2.5. Ultrafiltration Performance of the CA–g–AN Modified Membrane

A measurement of the water flux and rejection of the BSA solution was conducted to evaluate the ultrafiltration performance of the membrane. As shown in Figure 7a, by using CA–g–AN to modify the membranes, it was demonstrated that these membranes had a higher permeability than the pure PES membranes and that their permeability increased over time as the CA–g–AN content increased. As a result of the improved hydrophilicity and membrane structure, the permeability improved. The hydroxyl, carboxyl, and amide groups in CA–g–AN could modify the surface of the membrane, forming a layer of hydrophilic molecular brushes that enhanced the water–membrane interaction. In addition, the enlarged membrane pores reduced the path of water through the membrane and prevented obstructions. According to the results, the low concentration of CA–g–AN appeared to be more compatible with the membrane and, as a result, filled the membrane surface. This led to the BSA rejection of a PES/0.10–CA–g–AN increase in the first place. However, CA–g–AN simultaneously altered the cross-sectional structure of the membranes, resulting in broader pores. In this manner, the rejection of BSA decreased with an increase in CA–g–AN. As a result of the CA–g–AN modification of the membranes (Figure 7c), a significant increase in the flux recovery ratio (*FRR*) was observed with the increase in the CA–g–AN content, demonstrating that CA–g–AN significantly contributed to the antifouling properties of the PES membrane. According to Figure 7d, a decrease in the irreversible fouling ratio (*R_ir_*) of the modified membranes was observed and an increase in the reversible fouling ratio (*R_r_*) was observed. As a result of the study, it was found that the membranes modified with CA–g–AN possessed certain antifouling properties as well as the ability to convert a portion of irreversible fouling into reversible fouling.

## 3. Materials and Methods

### 3.1. Materials

In this work, all reagents used in the experiments were of an analytical grade without further purification. Polyethersulfone (PES, E6020P) was provided by BASF (Ludwigshafen, Germany). Polyvinylpyrrolidone (PVP), citric acid (CA), and N,N-dimethylacetamide (DMAc) were purchased from Tianjin Damao Co., Ltd (Tianjin, China). N-(3-dimethylaminopropyl)-N-ethylcarbodiimide hydrochloride (EDC) and N-hydroxysuccinimide (NHS) were supplied by Shanghai Aladin Biochemical Technology Co., Ltd. (Shanghai, China). N,N-dimethylformamide (DMF) was purchased from Shanghai Macklin Biochemical Technology Co., Ltd. (Shanghai, China).

### 3.2. Synthesis of CA–g–AN

CA–g–AN was synthesized from the amide bonds between the carboxyl groups of CA and the amino groups of AN. Briefly, CA (0.1 M) was slowly added to 10 mL DMF in a 50 mL round-bottomed flask. A certain molar ratio of EDC and NHS was added to the reaction vessel and the mixture was continuously stirred for 12 h at room temperature. AN (0.3 M) was slowly added to the mixture, and the stirring was continued for another 18 h. Ultrapure water was then added to the reaction vessel until the end of the reaction. The resulting mixture was stirred at room temperature for 12 h to obtain the precipitated substance. The substance obtained after repeated washing was further washed with copious amounts of deionized water (DI) and dried in a vacuum to obtain the yellow-brown powder of CA–g–AN.

### 3.3. Preparation of the Membranes

The membranes were prepared by non-solvent induced phase inversion (NIPS). Typically, PES, PVP, CA, or CA–g–AN were dissolved in DMAc at 60 °C for 6 h. After degassing at room temperature for 24 h, the casting solution was casted onto a glass plate using an MSK-AFA-III Automatic Thick Film Coater by a casting knife with a gap of 300 μm and a speed of 20 mm/s. After waiting in air for 30 s, the glass plate was subsequently immersed into pure water for 10 min at 25 ± 1 °C. Finally, the prepared membranes were kept in pure water for 24 h to remove the residual solvent, thus obtaining the final sheet. The formulation of the casting solution is shown in Table 2.

### 3.4. Characterization of CA–g–AN

A Fourier transform infrared spectrometer (FT-IR, Nicolet iS50, Thermo Fisher Scientific, Waltham, MA, USA) was used to evaluate the bonding pattern of CA–g–AN. The combination of CA–g–AN was further verified by a thermogravimetric analysis (TGA, TGA/DSC 3+, Mettler-Toledo (Schweiz) GmbH, Greifensee, Switzerland) between 35–800 °C at a heating rate of 10 °C/min under an air environment. The initial changes in the elemental composition of all the as-synthesized materials were confirmed by an X-ray photoelectron spectrometer (XPS, Escalab 250Xi, Thermo Fisher Scientific).

### 3.5. Characterization of the Membranes

The hydrophilicity of the membranes was analyzed by a contact angle meter (SDC-100s, SDC Technologies, Shanghai, China) and an atomic force microscope (AFM, Multimode Nanoscope-V, Bruker, Billerica, MA, USA). An XPS analysis (Escalab 250Xi) of the membrane surface was performed to analyze the surface roughness and elements of the membranes. The cross-section morphology of the membranes was evaluated by a field-emission scanning electron microscope (SEM, Phenom XL, Thermo Fisher Scientific). The membrane samples were ruptured in liquid nitrogen.

### 3.6. Porosity and Mean Pore Size Analysis of the Membranes

The method of wet and dry gravimetry was used to measure the porosity and mean pore size of the membranes [47]. The mass of the wet membranes with an area of 4 cm^2^ was measured first; these were then dried for 12 h in an oven at 100 °C. The mass of the dry membranes was further measured. The porosity (*ε*) was calculated as (Equation (1)):(1)$$\varepsilon =\frac{W_{w}-W_{d}}{\rho_{w}A\delta }\times 100\%,$$
where *W_w_* and *W_d_* (*g*) are the wet and dry membrane weights, respectively; *ρ_w_* (g·cm^−3^) is the density of water; *A* (cm^2^) is the membrane area; and *δ* (cm) is the membrane thickness.

The Guerout–Elford–Ferry equation [47] was used to calculate the mean pore size (*r_m_*) of the membrane (Equation (2)).
(2)$$r_{m}=\sqrt{\frac{\left(2.9-1.75\varepsilon \right)8\eta \delta Q}{\varepsilon A\Delta P}},$$
where *η* (8.9 × 10^−4^ Pa·s) is the viscosity of water, *Q* (m^3^·s^−1^) is the volume of permeated water, and ∆*P* (0.15 MPa) is the operating pressure.

### 3.7. Ultrafiltration Experiment of the Membranes

A dead-end filter module with an effective filtration area of 38.465 cm^2^ was used to test the flux and BSA rejection rate at 0.15 MPa. The membrane samples were pre-pressed with deionized water at 0.15 MPa for 30 min.

The flux of the pure water of the membranes was calculated as (Equation (3)):(3)$$J=\frac{V}{A\times T\prime }$$
where *V* (L) is the volume of the permeated pure water, *A* (m^2^) is the effective membrane area, and *T* (h) is the permeation time.

The pure water was substituted with 1 g·L^−1^ BSA solution. The BSA rejection ratio (*R*) was calculated as (Equation (4)):(4)$$R\left(\%\right)=\left(1-\frac{C_{p}}{C_{f}}\right)\times 100\%,$$
where *C_f_* is the concentration of BSA in the stock solution and *C_p_* is the concentration of BSA in the permeate. A UV-Vis INESA N4 (China) was used to measure the protein concentration at 280 nm.

The membranes were cleaned with DI water for 20 min, and were then used to test the water flux again. The flux recovery ratio (*FRR*), total fouling ratio (*R_t_*), reversible fouling ratio (*R_r_*), and irreversible fouling ratio (*R_ir_*) were calculated based on Equations (5)–(8):(5)$$FRR\left(\%\right)=\frac{J_{w1}}{J_{w2}}\times 100\%,$$
(6)$$R_{t}\left(\%\right)=\left(\frac{J_{W1}-J_{P}}{J_{W1}}\right)\times 100\%,$$
(7)$$R_{r}\left(\%\right)=\left(\frac{J_{W2}-J_{P}}{J_{W1}}\right)\times 100\%,$$
(8)$$R_{ir}{=R}_{t}-R_{r},$$
where *J_w1_* and *J_w2_* are the water fluxes before and after the BSA solution ultrafiltration, and the *J_p_* is the flux of the BSA solution.

## 4. Conclusions

In this work, based on CA and AN, an amphiphilic graft polymer CA–g–AN was synthesized by amide and hydrogen bonding. By filling and reducing the surface roughness of the membranes, the amphiphilic CA–g–AN became more compatibility. Moreover, the hydrophilicity of the membrane was enhanced by the hydrophilic groups (–COOH, –OH, and amide groups) of CA on CA–g–AN, resulting in a reduction in the WCA of 76.43° from 86.19°. Adding CA–g–AN also had a significant impact on the membrane structure, resulting in improved hydrophilicity; this promoted a higher water permeability (290.13 L·m^−2^·h^−1^). In addition, the *FRR* of the modified membranes reached 86.26%, demonstrating a better antifouling property than the pure PES membrane. This novel strategy may improve the engineering of antifouling membranes as a result.

## Figures and Tables

**Figure 1 molecules-28-01936-f001:**
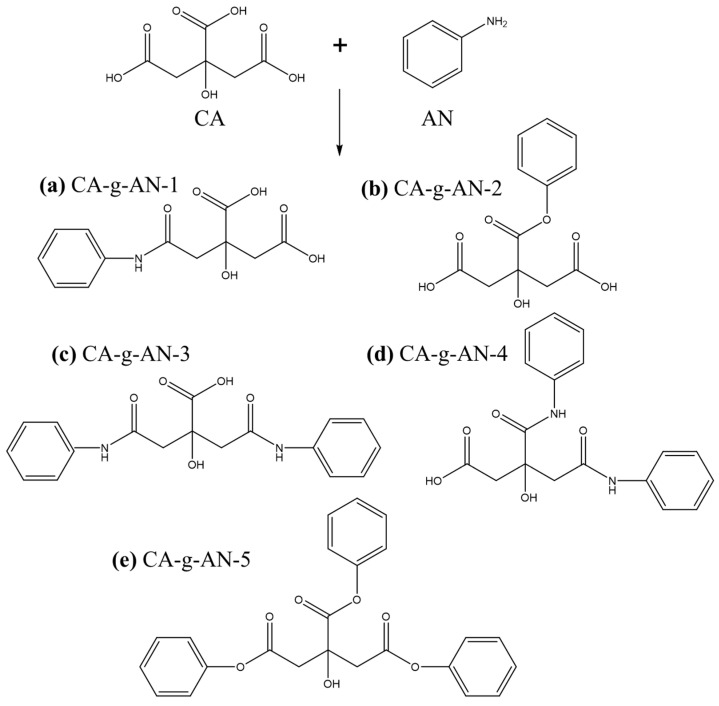
The possibility of the molecular structure of CA–g–AN. (**a**) CA–g–AN–1, (**b**) CA–g–AN–2, (**c**) CA–g–AN–3, (**d**) CA–g–AN–1 and (**e**) CA–g–AN–5.

**Figure 2 molecules-28-01936-f002:**
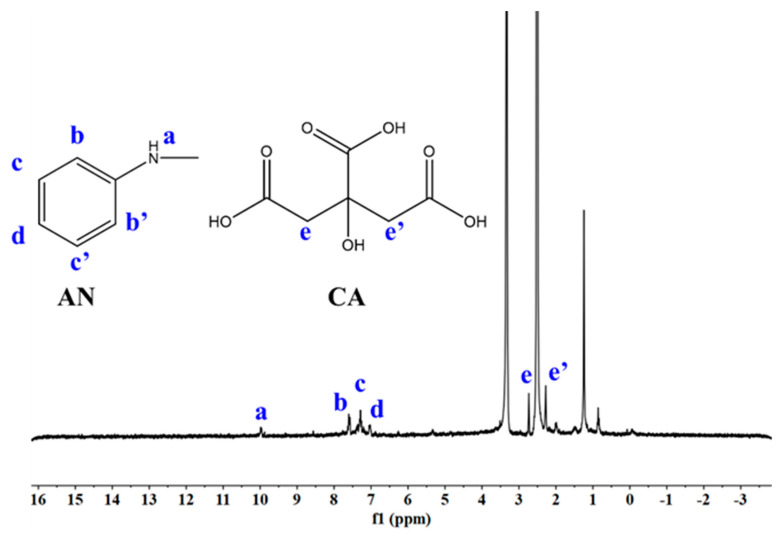
^1^H NMR spectrum of CA–g–AN.

**Figure 3 molecules-28-01936-f003:**
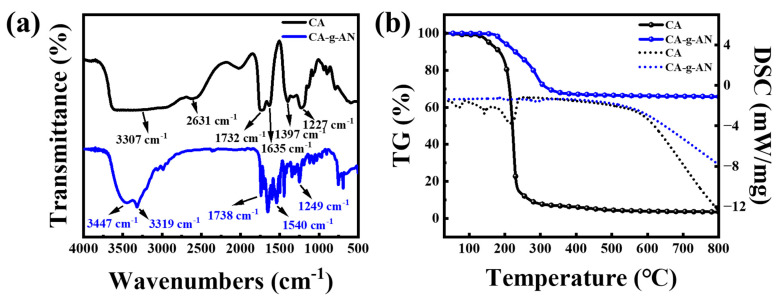
(**a**) FT-IR spectra and (**b**) thermal stability of CA and CA–g–AN.

**Figure 4 molecules-28-01936-f004:**
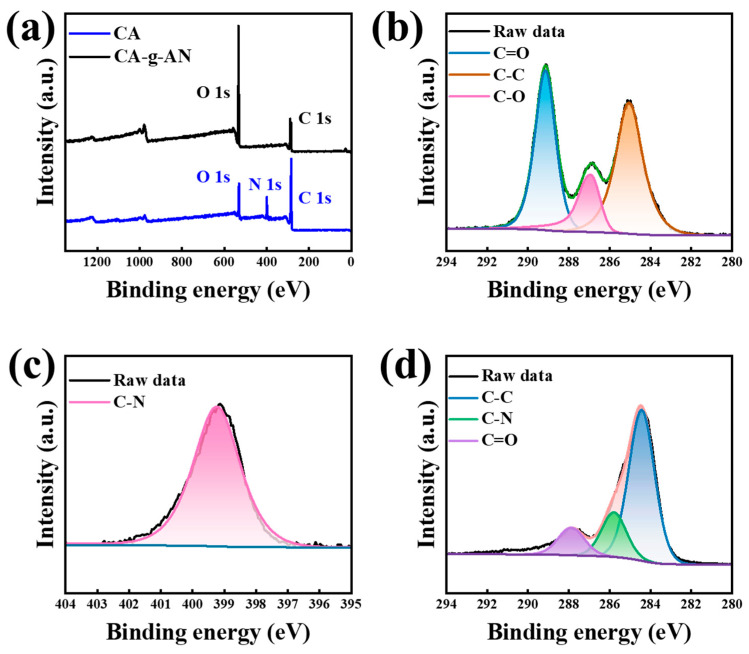
(**a**) XPS spectra of CA and CA–g–AN; high-resolution XPS (**b**) C 1s of CA, (**c**) N 1s, and (**d**) C 1s of CA–g–AN.

**Figure 5 molecules-28-01936-f005:**
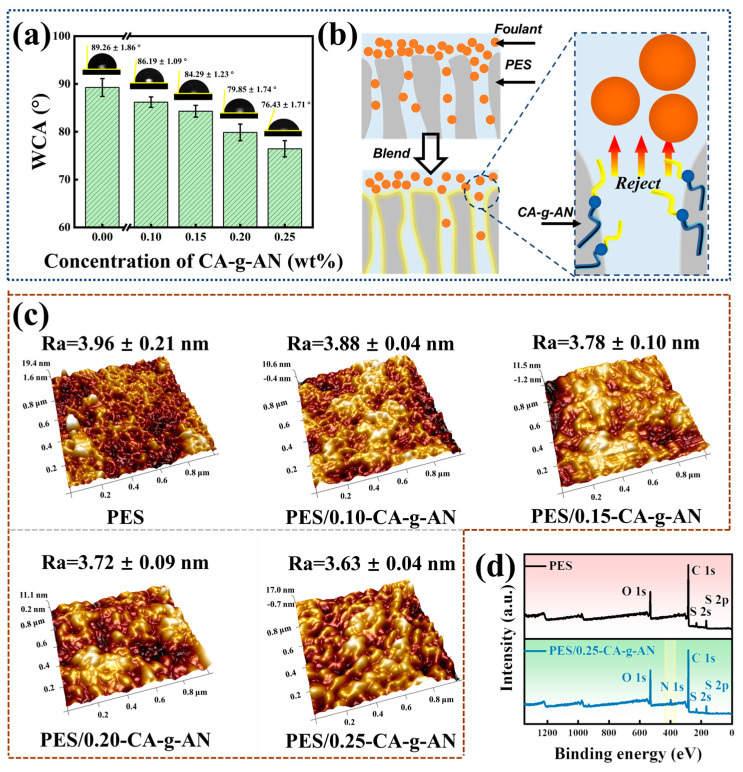
(**a**) The water contact angle of the membranes; (**b**) schematic diagram of antifouling mechanisms; (**c**) AFM images and surface roughness measurement values of the membranes; (**d**) XPS analysis of PES/0.25–CA–g–AN membrane surface.

**Figure 6 molecules-28-01936-f006:**
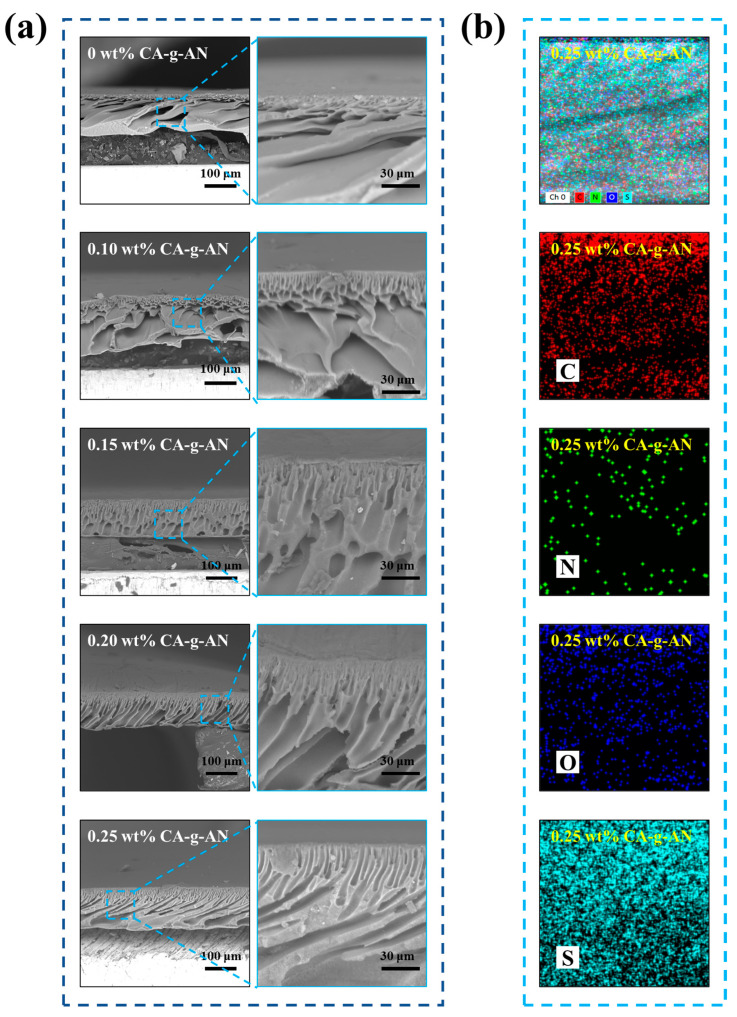
(**a**) The cross-section morphology of PES, PES/0.10–CA–g–AN, PES/0.15–CA–g–AN, PES/0.20–CA–g–AN, and PES/0.25–CA–g–AN membranes; (**b**) the distribution of C, N, O, and S elements on the PES/0.25–CA–g–AN membrane.

**Figure 7 molecules-28-01936-f007:**
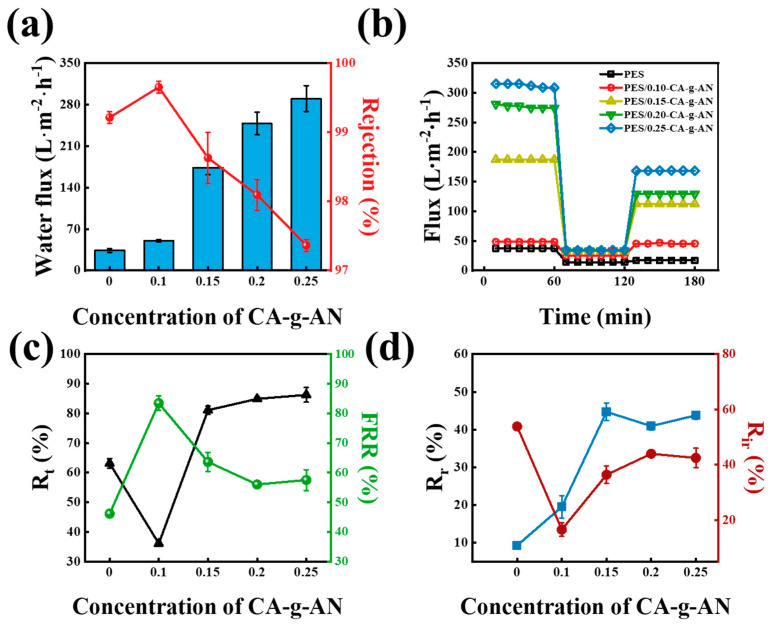
(**a**) Pure water flux and BSA rejection ratios; (**b**) time-dependent flux; (**c**) total fouling ratio (*R_t_*) and flux recovery ratio (*FRR*); (**d**) fouling ratio of PES, PES/0.10–CA–g–AN, PES/0.15–CA–g–AN, PES/0.20–CA–g–AN, and PES/0.25–CA–g–AN membranes.

**Table 1 molecules-28-01936-t001:** Porosity and mean pore size of membranes.

Membrane Name	Porosity (%)	Mean Pore Size (nm)
PES	66.84 ± 2.60	47.07 ± 2.24
PES/0.10–CA–g–AN	70.83 ± 3.09	49.47 ± 0.89
PES/0.15–CA–g–AN	71.90 ± 9.10	77.52 ± 2.38
PES/0.20–CA–g–AN	73.46 ± 1.79	85.47 ± 3.30
PES/0.25–CA–g–AN	77.64 ± 9.19	98.22 ± 3.62

**Table 2 molecules-28-01936-t002:** The formulation of the casting solution.

Membrane Name	PES (wt%)	PVP (wt%)	CA–g–AN (wt%)	DMAc (wt%)
PES	18.00	1.00	0.00	81.00
PES/0.10–CA–g–AN	18.00	1.00	0.10	80.90
PES/0.15–CA–g–AN	18.00	1.00	0.15	80.85
PES/0.20–CA–g–AN	18.00	1.00	0.20	80.80
PES/0.25–CA–g–AN	18.00	1.00	0.25	80.75

## Data Availability

The data presented in this study are available on request from the corresponding author.

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
