# Peer review of "Amphiphilic Grafted Polymers Based on Citric Acid and Aniline Used to Enhance the Antifouling and Permeability Properties of PES Membranes"

_molecules, 2023, doi:10.3390/molecules28041936_

Round 1

Reviewer 1 Report

In this manuscript, CA-g-AN were synthesized and blended with PES to form the UF membranes with antifouling properties. However, this work brings no new idea interesting to the readers. Some discussions and conclusions presented in this work are questionable. The organization of manuscript should also be improved. Therefore, this work is suggested to be published in Molecules after major revision. The authors may wish to consider the following comments in a revised version:

1. In abstract, the author stated that “This study aims to develop a new strategy for improving the antifouling property…” However, to improve the antifouling property of UF membrane by blending with amphiphilic polymers is a strategy which have been investigated for about 20 years. Thus, to be honest, this study has not presented a new strategy for improving the antifouling property of UF membranes.

2. The introduction part is suggested to be well-organized to show the novelty of this work.

3. The molecular structure of CA-g-AN should be confirmed in this study. A CA molecular contains 3 carboxyl groups. Thus, for the reaction between CA and AN, 3 reaction products could be formed as one CA reacted with one AN, one CA reacted with two AN, and one CA reacted with three AN. In the Graphical Abstract, the author presented the reaction product with the structure of one CA reacted with three AN. However, in manuscript, the author stated that the CA-g-AN contained -OH and -COOH. These statements are contradiction. NMR analysis on CA-g-AN are suggested to confirmed the accurate molecular structure of CA-g-AN.

4. Line 187-189, “The reason is that the modification additive, amphiphilic CA-g-AN, is more compatible with the membrane matrix in the blend process, which would reduce the roughness of the membrane surfaces, promoting the membrane surface smoother” As the author said, improving the compatibility between PES and the additives, the surface roughness could be reduced. Thus, PES should show the most smooth surface, since the compatibility between PES and PES must high than the one between PES and CA-g-AN. What about the surface roughness of the other modified UF membranes?

5. To study the effect of CA-g-AN on the membrane microstructure, the surface SEM images of the membranes should also be presented.

6. Line 239-240, “it appears that the low concentration of CA-g-AN is more compatible with the membrane, and that it fills the membrane surface, reducing the membrane surface pore size, so the BSA rejection of the PES/0.10-CA-g-AN increases firstly”. In Table 1 the pore size of PES/0.10-CA-g-AN was larger than PES. These two results were contradiction.

7. What about the MWCO of the prepared UF membranes?

8. What about the stability of the CA-g-AN modified UF membranes? Could CA-g-AN leach out from the membranes during the filtration process?

Reviewer 2 Report

This study examines a novel amphiphilic CA-g-AN material, constructed by grafting citric acid to aniline, as a modified material for improving the hydrophilicity of a PES membrane. Overall, the paper is well written and covers all requested aspects for the interest subject. I believe this study could be suitable for publication but will require minor revision before its consideration. The following minor aspects should be considered:

1.      Please correct Wang [24] to Wang et al. [24], and other such errors

2.      Line 276 “Deionized” with a lowercase letter

3.      Probably you should correct the names of axes in Figure 5c, due to the fact that you noted “FRR of the modified membranes can reach to 86.26%”. This statement does not correspond with data in Figure 5c. Please check the Figure 5c. In connection with this comment if Line 245 “a significant increase in total fouling ratio (Rt)…with the increase in CA-g-AN content” is correct?

4.      When you give an abbreviation for the first time, explain it then.

Round 2

Reviewer 1 Report

In this revision, most of the comments have been responded. The current version of this manuscript is suggested to be published in Molecules.